# Anthropogenic climate change has altered primary productivity in Lake Superior

M.D. O'Beirne[1,†], J.P. Werne[1,†], R.E. Hecky[1,2], T.C. Johnson[1,3], S. Katsev[1,4] & E.D. Reavie[5]

Anthropogenic climate change has the potential to alter many facets of Earth's freshwater resources, especially lacustrine ecosystems. The effects of anthropogenic changes in Lake Superior, which is Earth's largest freshwater lake by area, are not well documented (spatially or temporally) and predicted future states in response to climate change vary. Here we show that Lake Superior experienced a slow, steady increase in production throughout the Holocene using (paleo)productivity proxies in lacustrine sediments to reconstruct past changes in primary production. Furthermore, data from the last century indicate a rapid increase in primary production, which we attribute to increasing surface water temperatures and longer seasonal stratification related to longer ice-free periods in Lake Superior due to anthropogenic climate warming. These observations demonstrate that anthropogenic effects have become a prominent influence on one of Earth's largest, most pristine lacustrine ecosystems.

[1] Large Lakes Observatory (LLO), University of Minnesota Duluth (UMD), Duluth, Minnesota 55812, USA. [2] Department of Biology, UMD, Duluth, Minnesota 55812, USA. [3] Department of Earth and Environmental Sciences, UMD, Duluth, Minnesota 55812, USA. [4] Department of Physics, UMD, Duluth, Minnesota 55812, USA. [5] Center for Water and the Environment, Natural Resources Research Institute, UMD, Duluth, Minnesota 55811, USA. † Present address: Department of Geology and Environmental Science, University of Pittsburgh, Pittsburgh, Pennsylvania 15260, USA. Correspondence and requests for materials should be addressed to M.D.O. (email: mdobeirne@pitt.edu).

Lakes constitute ca. 2.8% of Earth's land surface area[1], yet they serve as critical resources for society by providing water for drinking, hygiene, industry, power generation and recreation. Lakes are also hot spots of biodiversity and are often sensitive indicators of environmental changes, both regional and global. The interconnectedness of lakes to their surrounding environments—the habitats they provide and biodiversity they hold, along with society's heavy reliance on their resources makes managing any resultant effects of environmental change important. To assess and subsequently mitigate potential impacts of current environmental change (that is, climate change) on ecosystem function[2,3], it is essential to know how lakes have responded to past environmental changes. This is primarily accomplished through the reconstruction of paleolimnological trends using lake sediment core records. This is especially useful in systems where historical records are lacking.

Lake Superior is Earth's largest freshwater lake by surface area and third largest by volume. Despite its prominence, both regionally and globally, documenting the duration and extent of environmental change in Lake Superior is often difficult because of the paucity of historical data, notably for primary production (PP). While details of the lake's primary productivity rates and nutrient cycles are still debated[4–6], isotopic data have demonstrated that autochthonous PP must dominate carbon inputs to Lake Superior[6].

To augment historical records and establish a baseline of PP in Lake Superior, we present a ca. 9000-year (paleo)productivity record from three piston and corresponding gravity sediment cores taken from across the lake basin (Fig. 1). (Paleo)productivity proxy data[7] (total organic carbon—TOC, atomic ratio of organic carbon to nitrogen—$C_{org}$:N, and the stable isotope composition of organic carbon—$\delta^{13}C_{org}$) from these cores provide a historic baseline for our assessment of current productivity trends inferred from eight sediment multicores (Fig. 1). The use of such proxies to reconstruct PP is based on the premise that increases in lacustrine productivity should lead to increases in the deposition of TOC along with changes in $\delta^{13}C_{org}$. In general, the relationship between the $\delta^{13}C$ of primary photosynthate and the extent of PP reflects the response of isotopic fractionation during carbon fixation to variations in the concentration of aqueous $CO_2$ (ref. 8). Essentially, $^{12}C$ is more energetically favourable in the assimilation of $CO_2$ during photosynthesis. As photosynthesis increases, reserves of $^{12}C$ within the water column become depleted and discrimination of $^{13}C$ is reduced. As a result, $\delta^{13}C_{org}$ values are typically $^{13}C$-enriched (depleted) with increasing (decreasing) PP. Thus, changes in TOC and $\delta^{13}C_{org}$ can be used as a proxy for changes in lacustrine (paleo)productivity.

The eight sediment multicores were sampled at high resolution and encompass the last 200 years—the time of European habitation of the region and period of rapid anthropogenic climate change. This is the first sedimentary multi-site record of geochemically inferred PP increases in Lake Superior spanning the Holocene and which also demonstrates that abrupt increases in PP within the last century can be reasonably explained by the effects of anthropogenic climate warming.

## Results

**Overview of the 9000 year paleoproductivity record.** Although post-glacial TOC concentrations are low ($< 1\%$), by ca. 7500 cal BP TOC and $\delta^{13}C_{org}$ values begin to increase concurrently (Fig. 2), which we attribute to increasing PP within Lake Superior. This designation is consistent with a previous study[9], although that sedimentary record reports no data from 3000 cal BP – present. Values of atomic $C_{org}$:N ratios in the cores presented are less than 10 (Fig. 2d), which indicates sediments dominated by algal sourced organic matter (OM)[7]. Gradual, long-term increases in TOC and $C_{org}$:N suggest the possibility of slightly increased terrestrial input to the lake basin; however, a cross-plot of $C_{org}$:N versus $\delta^{13}C_{org}$ (Fig. 2d) demonstrates that autochthonous PP is the dominant influence on the sedimentary organic record throughout the entire post-glacial period (ca. 8500 cal BP – present). Although soil OM adsorbed onto fine-grained clays can also exhibit $C_{org}$:N ratios less than 10 (refs 10,11), the remarkable coherence among cores regardless of proximity to shore (the source of clay-derived soil OM) is more suggestive of algal sourced OM buried in the sediments.

**The most recent increases in TOC and $\delta^{13}C_{org}$.** The most salient feature in all sediment cores is the abrupt increase in bulk productivity proxy indicators over the last century (Fig. 2). The increase occurs more rapidly than at any other time period in the record(s) presented (Figs 2 and 3). The general pattern observed in the bulk sediment carbon isotope values ($\delta^{13}C_{org}$) for all cores is a period of $^{13}C$-enrichment (ca. 2‰) beginning near 1900AD and continuing until present day. The timing of the increase in both TOC and $\delta^{13}C_{org}$ varies among the core sites. Overall, the cores nearer to shore and in shallower water depths show slightly later increases than those that are farther off-shore and in mid-to-deep water depths.

**Determining the effects of diagenesis.** To account for the role of diagenesis in altering the sedimentary record (that is, TOC mineralization) we constructed steady-state diagenetic models that were then compared with observed profiles. The modelled profiles are concordant with our measured TOC profiles (Fig. 4). Thus, TOC concentrations, when considered alone, must be interpreted with caution (see Supplementary Discussion). In Lake Ontario, post-burial diagenesis reduced the organic carbon content within sediment cores taken 6 years apart, but it did not significantly alter the isotopic ($\delta^{13}C_{org}$) composition[12]. Similar trends were observed in a pair of multicores taken 6 years apart from western Lake Superior[13], supporting the interpretation that $\delta^{13}C_{org}$ values reliably record the primary signal (changes in PP) despite OM loss. Consequently, our focus turns to interpretation of the $\delta^{13}C_{org}$ record. Additional studies[7,14] have similarly shown that isotopic discrimination associated with (re)mineralization has a negligible influence on $\delta^{13}C_{org}$ within oxic, low organic content sediments like Lake Superior. Therefore, diagenesis provides an unconvincing explanation for the overall $^{13}C$-enrichment of OM observed and certainly not the exceptional $^{13}C$-enrichment documented in the last century in Lake Superior.

## Discussion

Gradual increases in $\delta^{13}C_{org}$ until ca. 1900AD are likely due to the natural, low-level delivery of Fe and P from weathering of the iron-rich bedrock within the surrounding watershed. Lake Superior is presently co-limited by Fe and P (refs 15,16). Over time a steady, low-level delivery of both elements to the lake would slowly increase the amount of each to the recycled pool (Fe and P are rapidly recycled in the water column) and allow for a sustained increase in PP. Increases in nitrate concentrations likely helped to fuel initial PP in the lake; however, the continuous build-up throughout the last century and extremely high N:P ratios at present[17] suggest that the amount of available N is not what is limiting PP in Lake Superior. Increasing $\delta^{13}C_{org}$ values after 1900AD are consistent with increasing autochthonous PP (Fig. 3) and coincide with increased anthropogenic disturbance (increasing nutrient flux) within the Lake Superior watershed[13,18]. Beginning in late 1970AD strict

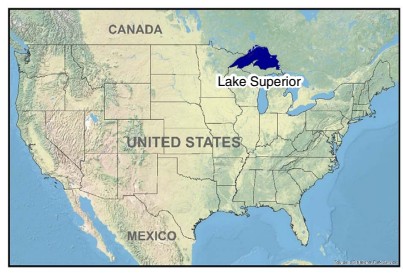
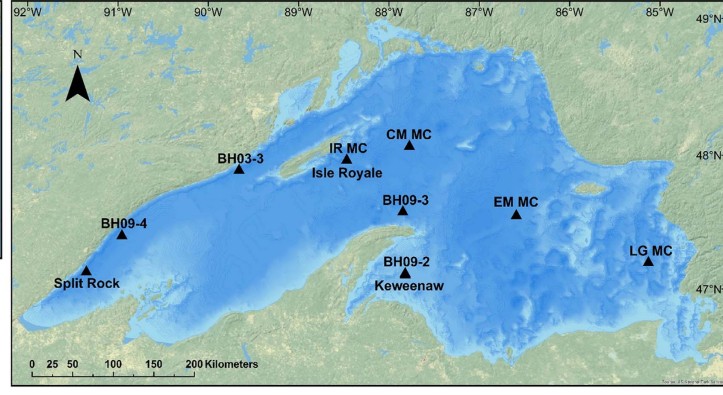

**Figure 1 | Map of Lake Superior with sediment core locations.** BH03-3, BH09-2, BH09-3, BH09-4, IR MC, CM MC, EM MC and LG MC are multicores. Split Rock, Isle Royale and Keweenaw are piston/gravity cores.

regulation on anthropogenic nutrient inputs to Lake Superior were initiated[19]. Thus, we might expect $\delta^{13}C_{org}$ values to decrease or remain stable. The causes for the observed lake-wide productivity increase in the most recent century are enigmatic and possible explanations are discussed below.

Causes for the $^{13}C$-enrichment of bulk sedimentary OM include changes in environmental and/or physiological factors that may alter the isotopic composition of photosynthetic organisms, thus altering the isotopic signature of bulk OM without increasing or decreasing PP. Such processes might include changes in the source of inorganic carbon, isotope effects associated with carbon assimilation and metabolism, a shift in the phytoplankton community structure to one with a greater abundance of $^{13}C$-rich organisms (for example, diatoms[20]), and/or temperature effects.

Arguably, the two major sources of inorganic carbon to Lake Superior are atmospheric $CO_2$ influx (owing to its large surface area; $\sim 82,000\,km^2$) and dissolved inorganic carbon (DIC) from rivers and streams. Lake $\Delta^{14}C_{DIC}$ tracks atmospheric $\Delta^{14}C_{CO2}$ values rather than riverine $\Delta^{14}C_{DIC}$ values[21]. Therefore, changes in watershed factors such as soil respiration that could influence lake DIC $\delta^{13}C$ values through time are likely negligible.

Limited monitoring data[22,23] suggest there has been little variation in the relative composition of algal divisions in Lake Superior; thus it is assumed that the mechanism(s) of carbon assimilation and metabolism have not changed significantly over the last century. Additionally, while diatoms make up a large proportion of the phytoplankton community in Lake Superior[22], it is not likely that their overall percentage is the sole factor producing the $^{13}C$-enrichment of bulk sedimentary OM between 1900 and present, as there is no (or negative) correlation between $\delta^{13}C_{org}$ and diatom biovolume measurements (see Supplementary Discussion). Furthermore, the subtle changes in algal divisions that have been documented are likely the result of warming water temperatures and accompanied by changes in water quality[22].

Temperature effects could contribute to shifts in the isotopic composition of primary photosynthate. Over the last century, Lake Superior has warmed considerably (ca. 3.5 °C during the summer) with shifts to longer ice-free and stratified periods[24,25]. Increasing water temperatures would lead to the evasion of isotopically light $CO_{2(aq)}$, causing an overall $^{13}C$-enrichment in DIC[26,27]. Assuming a conservative 0.1‰ $^{13}C$-enrichment in DIC per °C increase in temperature[28] the maximum $^{13}C$-enrichment in DIC accounted for by increasing surface water temperature is 0.35‰, which is considerably less than the ca. 2‰ enrichment we observe. Lengthened periods of thermal stratification can also induce changes in DIC isotope compositions by limiting mixing of the water column and therefore limiting upwelling of

$^{13}C$-depleted respired DIC produced at depth[29]. The upper water column (photic zone) DIC then becomes $^{13}C$-enriched as photosynthesis occurs throughout the stratified season[5]. Consequently, OM produced via photosynthesis becomes increasingly $^{13}C$-enriched as the stratified season progresses without increasing cell-specific rates of PP.

Water column measurements of PP in Lake Superior, although limited, vary in magnitude and display no steady increase (ref. 4 and others therein) in contrast to what we present here. We posit that this discrepancy stems from the fact that traditional water column measurements have been few and relatively infrequent[4], whereas the sedimentary record is an integrated signal of the entire year—capturing total annual patterns in PP. Traditionally, water column measurements are taken once in the spring and once in the summer at one or two locations, but not every year. No measurements of PP have been taken during ice-covered months, which omits a potentially substantial contribution to total annual PP (ref. 30 and others therein) when using traditional measurements alone. That said, the direct measurements of PP in the ice-free season on Lake Superior display little or no seasonal variation[4]. At any point in time primary productivity rates may remain constant but if the growing season becomes longer, net PP and subsequent export to the sediments can increase. Owing to this fact and others discussed above, we interpret the observed increases in sedimentary TOC and $\delta^{13}C_{org}$ values as increases in export production (that is, an increase in the amount of OM produced by PP that is not recycled) due to the increasing length of the growing season (that is, longer ice-free and stratified periods associated with increasing surface water temperatures[24,25]). Indeed, a recent modelling study[31] shows increasing total annual gross primary productivity with increasing temperature and decreasing ice-cover (simulation period 1985–2008) in Lake Superior.

There is some variability in the timing of the most recent increase in PP among the multicores presented here. Owing to the heterogeneous nature of the lake (and all large lakes), one may not expect a simultaneous basin-wide response. Lake surface water temperature is not constant across the spatial extent of the lake, but varies considerably based on factors such as water depth, physical circulation patterns and ice-cover. Furthermore, the extent of warming and decrease in ice cover in recent years have both varied across the lake basin (ref. 32 and others therein). Despite the timing of increases in TOC and $\delta^{13}C_{org}$ seen in the individual multicores presented here, the overwhelming response is that of increased lake-wide PP which coincides with the effects of increased temperatures (throughout the last century, but most markedly in the last 30 years[24,25]) in Lake Superior.

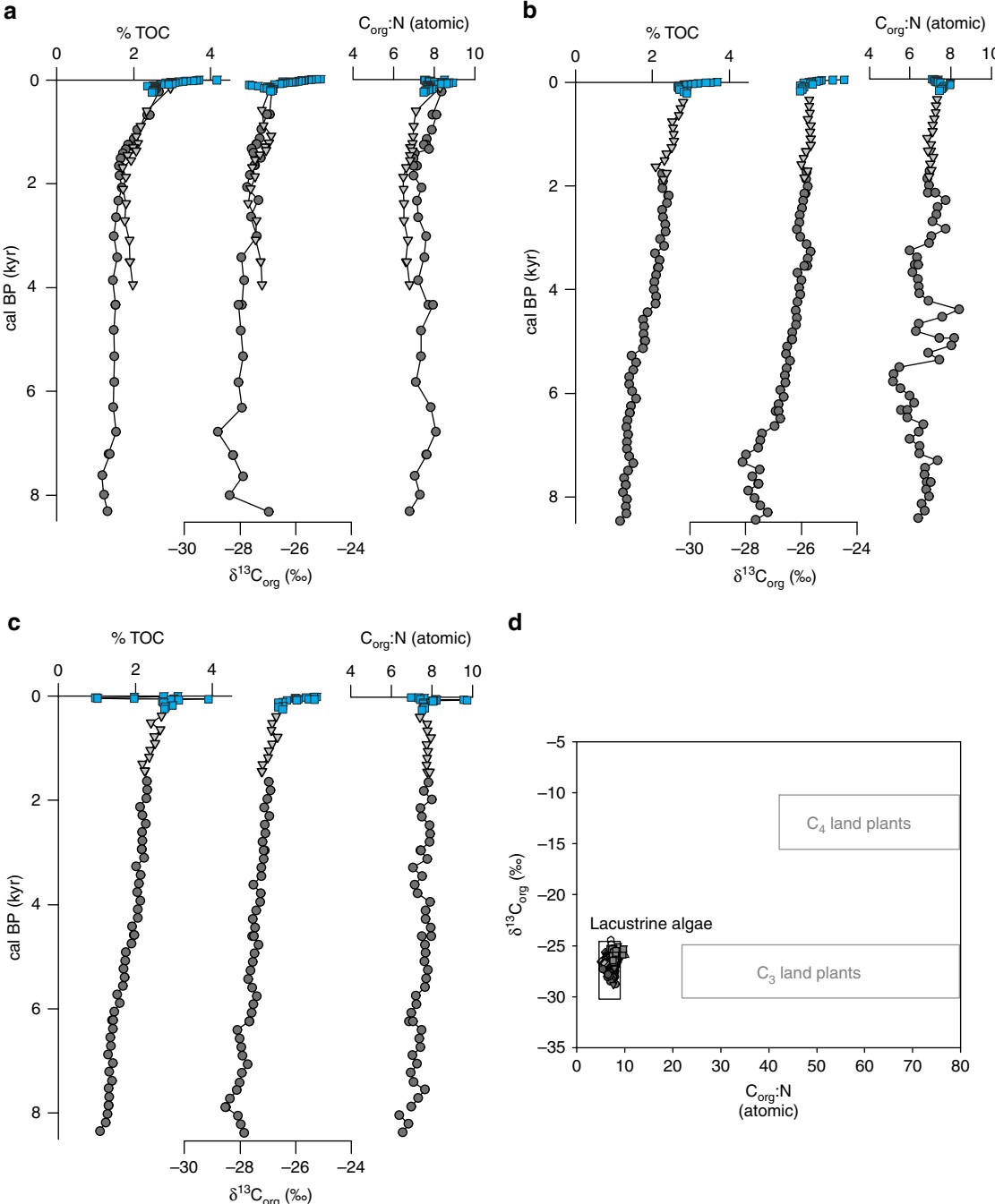

**Figure 2 | Paleoproductivity proxies plotted versus calendar years before 1950 for three locations in Lake Superior.** (**a**) Isle Royale; (**b**) Keweenaw; (**c**) Split Rock; (**d**) cross-plot of $C_{org}$:N versus $\delta^{13}C_{org}$ showing values plot within the lacustrine algal range. Dark grey circles are data from the piston cores, light grey triangles are data from the gravity cores, blue squares are data from the multicores. %TOC is plotted as a wt./wt. ratio of dry sediment, $C_{org}$:N as atomic ratios and $\delta^{13}C_{org}$ as per mil (‰) relative to the standard—Vienna PeeDee Belemnite (VPDB).

It is clear from data presented here that the increase in PP within the last century has not been experienced previously in Lake Superior and that past changes in climate regime(s), including abrupt climate changes, were not as influential as those today. For example, a ca. 200-year period of cooling that could have reduced algal productivity (the so-called 8.2 ka event[33]) is not apparent. Likewise, there is no evidence of either the Medieval Climate Anomaly (950–1250AD (ref. 34); exhibiting the warmest 50-year period of Northern Hemisphere mean annual temperature before 1900AD (ref. 34)), or the Little Ice Age (1400–1900AD (ref. 34)). Sampling resolution of the piston cores

is approximately every 100 years—which should capture such events if present. Additionally, warmer mid-Holocene regional summer air temperatures were offset by cooler winter air temperatures[35], presumably resulting in greater ice-cover. In contrast, the rate of anthropogenic warming is much faster and acts as a positive feedback increasing in-lake temperatures faster than the surrounding air, thereby decreasing ice-cover and increasing the length of the stratified 'growing' season in Lake Superior[24,25]. Even though past climate perturbations are not apparent in the (paleo)productivity record presented here, it is likely that the rate at which the most recent environmental and

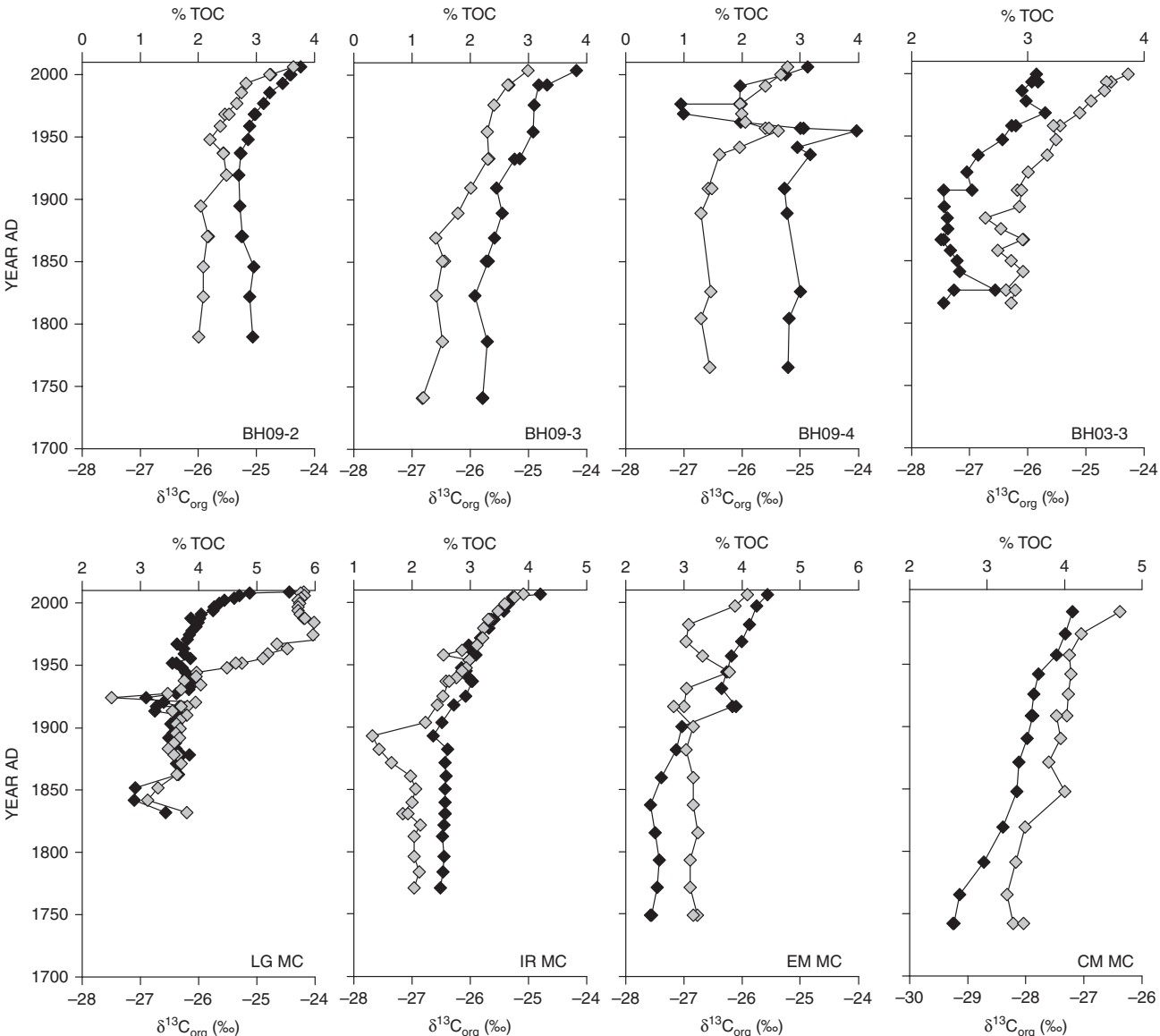

**Figure 3 | Temporal relationship between %TOC and $\delta^{13}C_{org}$ in eight sediment multicores taken from across the Lake Superior basin.** Increases in both %TOC and $\delta^{13}C_{org}$ are indicative of increasing lake-wide primary production. *Note varying horizontal scales. %TOC is plotted as a wt./wt. ratio of dry sediment, and $\delta^{13}C_{org}$ as per mil (‰) relative to the standard—Vienna PeeDee Belemnite (VPDB).

climate changes have occurred has overcome the buffering capacity of Lake Superior's large volume, allowing for recent changes to become manifest as increases in productivity. These observations suggest that anthropogenic influences are becoming increasingly important and are of significance as the great lakes of the world are considered less susceptible to anthropogenic changes (climate or otherwise) than their smaller counterparts.

Postglacial sediments of three piston and corresponding gravity cores taken from across the Lake Superior basin provide a historic baseline of primary productivity characterized by a slow and steady increase in TOC abundance, as well as $\delta^{13}C_{org}$ throughout the past 9000 years. Analysis of eight sediment multicores sampled at high resolution encompassing the last 200 years reveal that twentieth century increases in PP are unprecedented during the Holocene. We contend that changes in sedimentary TOC and $\delta^{13}C_{org}$ values within the last century are explained most parsimoniously by increased autochthonous PP driven by rapid anthropogenic climate warming and attendant limnological effects, including the increasing length of the stratified season

and surface water temperatures brought about through a positive feedback related to the length of the ice-free period. Ultimately, these results have important implications for nutrient cycling and food web dynamics in not only Lake Superior, but also all global freshwater resources as climate change persists.

## Methods

**Sediment cores.** Three lake sediment cores were retrieved from Lake Superior using a Kullenberg piston corer aboard the R/V Blue Heron (Fig. 1 in text; latitude and longitude listed in Supplementary Table 1); one in 2002 (BH02-10P; Split Rock—SR), one in 2009 (BH09K-SUP09; Keweenaw—KW) and one in 2011 (BH11IR-SUP11; Isle Royale—IR). Gravity cores were also taken as trigger cores corresponding to each of the three piston cores, providing a more complete sediment core record. Eight sediment multicores, designed to capture the sediment water interface, were collected aboard the R/V Blue Heron and one aboard the R/V Lake Guardian from each basin of Lake Superior (Fig. 1 in text; latitude and longitude listed in Supplementary Table 2); one in 2003 (BH03-3), three in 2009 (BH09-2, BH09-3 and BH09-4) and four in 2010 (LG MC, IR MC, EM MC and CM MC). Core sites were identified using a Knudsen 12 kHz Hi-Res echo sounder and a CHIRP seismic reflection profiler onboard the R/V Blue Heron. The coring sites were chosen based on evidence of uninterrupted stratigraphy resulting from

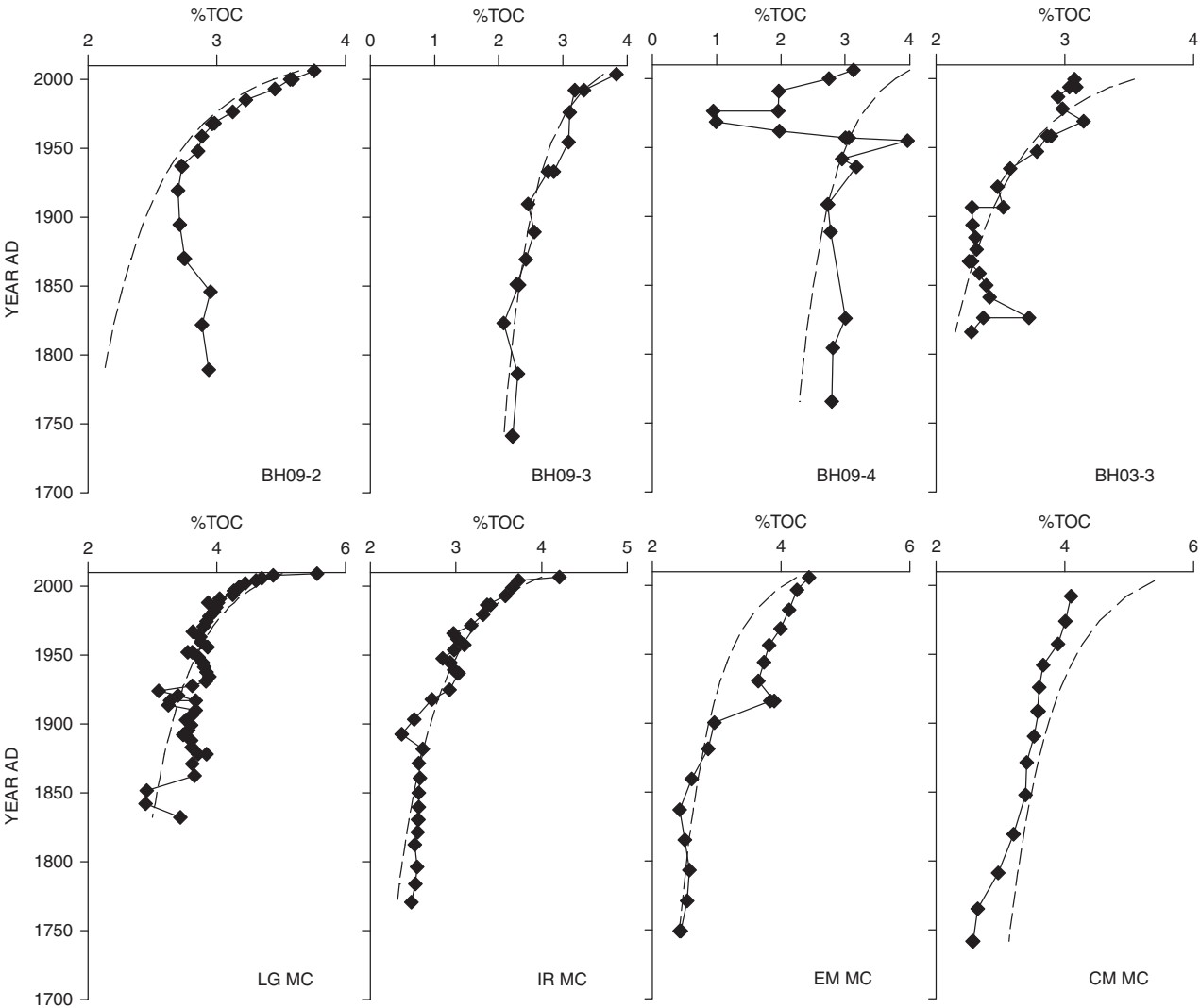

**Figure 4 | Profiles of measured and modelled %TOC.** Black diamonds are the measured total organic carbon values (%TOC). The dashed lines are modelled diagenetic proles of %TOC (see Supplementary Discussion). *Note varying horizontal scales. %TOC is plotted as a wt./wt. ratio of dry sediment.

continuous sedimentation to characterize representative locations in Lake Superior. The LG MC core was collected aboard the R/V Lake Guardian using an Ocean Instruments model 750 box corer (30 cm × 30 cm × 90 cm), from which two 6.5 cm internal diameter cylindrical cores were sub-sampled. The rest of the multicores were collected with an Ocean Instruments model MC-400 multi-corer (9.4 cm diameter) from water depths listed in Supplementary Table 2. Initial core description and splitting were performed at the National Lacustrine Core Repository (LacCore). Subsamples were taken from each multicore at 0.5 cm intervals to 10 cm and every 1 cm thereafter to the base of the core and from each piston and accompanying gravity (trigger) core at 10 cm intervals to the base of the core. Sediment samples were subsequently frozen and freeze-dried for geochemical analyses.

**Sample preparation.** Homogenized dry sediment samples of known weight (12–20 mg) were placed into silver (Ag) foil capsules in designated sample trays and 1 μl of nanopure filtered $H_2O$ was dispensed into each sample capsule. Sample trays were then placed in a desiccator, sans desiccant, with 12 M HCl for 8 h to remove inorganic carbon from the sediment (acid fumigation). Upon removal, samples were allowed to off-gas residual HCl while being dried on a hotplate set at 60 °C; drying time was on the order of 12 h. In order to ensure dryness, samples were additionally placed in an oven set at 60 °C overnight. Dried samples were then folded in tin (Sn) foil capsules.

**Elemental and isotopic analyses.** TOC and total nitrogen abundances of bulk sedimentary OM were analysed for weight percent C and N concurrently with isotope ratio determinations. All concentrations are presented for acid fumigated (decarbonated) samples, eliminating variability caused by carbonate concentrations within samples. $N_2$ and $CO_2$ peak areas (Isodat v3.0) were converted to weight

percent compositions using response factors generated from standards of known composition (acetanilide, caffeine, B-2153, B-2159 and urea), which were run between every ten samples. Analytical precision, based on replicate standard runs for bulk measurements, was typically better than ± 0.70% on average for carbon and nitrogen weight % values and better than ± 0.25‰ on average for carbon isotope values. Reproducibility between duplicate samples was better than ± 0.05% for both carbon and nitrogen weight % values; reproducibility of carbon isotopic values between duplicates was better than ± 0.07‰. Analyses were performed at the Large Lakes Observatory (LLO) Stable Isotope Lab using a Costech Elemental Analyzer coupled with a ThermoFinnigan Delta[plus]XP stable isotope ratio monitoring mass spectrometer (EA-IRMS). Every tenth sample was run in duplicate. Carbon isotopic ($δ^{13}C$) values are reported relative to Vienna Pee Dee Belemnite in conventional delta notation as per mil (‰) deviations.

**Suess correction.** To account for the change in the $δ^{13}C$ of atmospheric $CO_2$ from anthropogenic fossil fuel burning (the Suess effect) and consequent influence on bulk sediment $δ^{13}C$ values[13,36] all $δ^{13}C$ values were corrected using the equation from ref. 36, as it encompasses the entire time period spanned by the multicores (1700–2010 AD) in this study. For lack of quantitative information on the existence of reservoir effect in Lake Superior, the correction assumed no significant time delay for particulate organic carbon sedimentation.

**$^{210}Pb$ geochronology.** Multicores were subsampled at alternating 0.5 cm intervals to 10 cm depth and alternating 1 cm intervals from 10 to 20 cm. Samples at depth were included to determine the background (supported) $^{210}Pb$ activity. Unsupported $^{210}Pb$ activity should be negligible at depth, where sediment age is likely to exceed 150 years (>6 half-lives). The cores were not analysed for Cs-137 because the Cs-137 peak may fall at an age younger than the time of maximum

atmospheric concentration due to a lag in Cs-137 transport to its final depositional site via resuspension, transport and re-deposition in the Laurentian Great Lakes[37]. The [210]Pb analyses for the 2003 and 2009 cores were carried out by α-spectrometry in the Department of Soil Science, University of Manitoba, under the direction of Dr Paul Wilkinson, and at Flett Research Ltd. in Winnipeg, Manitoba under the direction of Dr Robert J. Flett for the 2010 cores. The [210]Pb analyses for LG MC were completed by Dr Dan Engstrom at the Science Museum of Minnesota St Croix Watershed Research Station. The age–depth relationships of the eight multicores were estimated from semi-log plots of excess [210]Pb activity versus accumulated sediment mass using previously described methods[38]. The slopes of the straight-line segments lower in the core are proportional to sediment mass accumulation rates (MARs), which are applied to arrive at sediment age at each horizon (Supplementary Figs 1 and 2). A bioturbation zone is apparent in the uppermost part of most cores and exhibits the steepest slope[38,39]. The MAR for this zone is assumed to be constant, and equal to that derived from the slope of the line segment immediately below the bioturbation zone. An excursion occurs in core BH09-4 between 3 and 7 cm and is attributed to the effects of taconite ore processing beginning in 1955, with court-ordered reductions in effect by 1980; therefore, MARs for this portion are assumed to be equivalent to the slope of the segment immediately above this disturbance, but below the bioturbation zone. The [210]Pb data reveal a bioturbated zone of 1.5 cm on average (Supplementary Table 2) in all cores, which is consistent with previous calculations[38,39]. Each zone is equivalent to 9 years in cores BH09-2, BH09-4 and BH03-3; 17, 4, 11, 13 and 18 years of sedimentation for cores BH09-3, LG MC, IR MC, EM MC and CM MC respectively. MARs in Lake Superior have not been constant over the periods of depositional history recorded in the cores—listed in Supplementary Table 2.

**Paleomagnetic secular variation.** Paleomagnetism is an often-used method for the age modelling of lacustrine sediments, especially where measurements of [14]C are unsuccessful. Radiocarbon dates are limited in Lake Superior, as macrofossils for dating are rarely present, and sediments of gravity and piston cores present in the current study are too old for [210]Pb dating. Thus, the alignment of paleomagnetic secular variation (PSV) records is the best method for dating Lake Superior sediments[40]. PSV records document regional variations in inclination and declination, which reflect variations in the Earth's magnetic field with time. Ferromagnetic grains (for example, magnetite) align themselves with the local magnetic field during or shortly after deposition, upon consolidation of the sediments the motion of these grains is constrained. Any magnetization acquired by the magnetic grains long after deposition is then removed in the laboratory by alternating field demagnetization at low fields. In contrast, the natural remanent magnetization removed at higher demagnetization fields is assumed to have resulted from burial in the presence of the local magnetic field[41]. The PSV records from the post-glacial sediments are 10–100 year averages of the regional field, due to a 1–2 cm bioturbation zone present in the top portions of Lake Superior sediments[38,39]. By correlating PSV records from Superior with PSV records from regional, well-dated sites (small lakes in the region), ages can be assigned to sediment cores[40]. natural remanent magnetization, providing inclination and declination data for correlative purposes, was completed in 2004 at Michigan Technological University—Earth Magnetism Laboratory under the direction of Dr Suzanne Beske-Diehl for piston core BH02-10P (Split Rock—SR) and by Dr Julie Bowles at the Institute for Rock Magnetism at the University of Minnesota for the piston cores BH09K-SUP09 (Keweenaw—KW) in 2009 and BH11IR-SUP11 (Isle Royale—IR) in 2012. PSV data for the three piston cores were compared to a previously dated core, LU83-8 (ref. 42). Site-specific age–depth profiles for each of the three piston cores were completed using LU83-8 ages and associated PSV features[40,42,43] (see Supplementary Fig. 3 and Supplementary Tables 3–5). For this study only features and associated ages of inclination were used in constructing the final age–depth model (regression line), due to the lack of features observable/present in the PSV declination data. Ages of inclination features apparent in each of the cores are presented as calibrated years before 1950. Correlative features for each of the three cores are listed in Supplementary Tables 3–5 and shown in Supplementary Fig. 3. The equation of the regression line (linear in SR and KW and fifth degree polynomial in IR) from the resulting plot of age versus depth from correlation with the previously dated core LU83-8 was then applied along the length of each of the cores providing the final age-depth assignments. Small amounts of Mazama ash, which is dated at 7700 cal YBP[44] were identified in the Isle Royale and Keweenaw piston cores[45], and included in the age models (Supplementary Fig. 4).

**Age–depth models for gravity cores.** Age/depth relationships for gravity cores were developed using linear extrapolation from the bottom of corresponding multicores (SR—BH09-4, KW—BH09-2 and IR—IR MC), which had previous [210]Pb age assignments, as described above. Extrapolation from the [210]Pb age model of the multicores to the gravity cores makes the assumption that the gravity cores begin where multicores end. Although not an ideal assumption, this is a valid approach as it provides a conservative age model, in addition to the fact that over penetration of the sediment/water interface is common when recovering both gravity and piston cores whereas multicores are designed to capture the sediment/water interface.

**Diagenetic modelling.** To account for the effects of diagenesis on the observed sedimentary TOC concentrations we constructed steady-state diagenetic models for each of the eight multicores, which we then compared to our measured values. Following refs 46–48, we used a reactive continuum model in which the effective bulk reactivity (k) of organic material in freshwater[46] as well as marine[47,48] environments decreases as a power law function of time (t) as in equation (1) where a and b are constants:

$$k(t) = bt^{-a} \qquad (1)$$

The rate of organic carbon mineralization R is defined in equation (2):

$$R = \frac{dC}{dt} = -k(t)C \qquad (2)$$

Equation (2) can be rewritten with substitution of the variable $k(t)$ from equation (1) and subsequently integrated to obtain the fraction of organic material that remains after a given amount of time. For $a \neq 1$, separation of variables and integration of equation (3) yields equation (4), where $C(t_0)$ is the concentration at the initial time $t_0$.

$$\frac{dC}{dt} = -bt^{-a}C \qquad (3)$$

$$C(t) = C(t_0) \exp\left(\frac{-b(t^{1-a} - t_0^{1-a})}{1-a}\right) \qquad (4)$$

The time $t_0$ corresponds to the age of OM at the sediment surface and effectively accounts for its ageing during settling through the water column and possibly for sediment resuspension[46]. For our purposes the initial concentration of organic carbon—$C(t_0)$—is set equal to the amount of organic carbon (%TOC) measured in the uppermost sediment interval with an offset considered (listed in Supplementary Table 2). The measured TOC concentration(s) in the uppermost sediment interval may not be an accurate representation of the concentration(s) at the very interface (mathematically defined), so the offset is used to correct for potential differences. $t_0$ is taken to correspond to the settling time of OM in the water column ($t_0 = 18$ years; estimated average for Lake Superior). The constants a and b were determined from the linear regression of $\log_{10} k$ versus $\log_{10} t$ from ref. 49 and are $a = -0.985$ and $b = 0.21$.

**Statistical analysis.** All statistical analyses reported here were performed using SigmaPlot v. 11.0.

**Generated map.** Lake Superior Core Locations [map]. Scale not given. Data layers: U.S. National Park Service: World Physical Map; Esri, TomTom, U.S. Department of Commerce, U.S. Census Bureau: USA States (Generalized); mbockenhauer; Michigan Department of Technology, Management and Budget: Great Lakes Bathymetry [computer files]. University of Pittsburgh, Pittsburgh, PA: Generated by Molly D. O'Beirne, 3 April 2017. Using: ArcGIS for Desktop [GIS]. Version 10.4. Redlands, CA: Esri, 2016.

**Data availability.** The datasets generated during the current study are available in the PANGAEA repository (doi:10.1594/PANGAEA.874731).

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

## Acknowledgements

We thank S. Grosshuesch and K. Dutta for analytical support, J. Halbur and L. Strzok for help with sample preparation, A. Breckenridge for help with paleomagnetic interpretation, J. Li for insights in modelling diagenesis, and the crew of the R/V Blue Heron and Lake Guardian for their expertise during core collection. Additional thanks go to Sue Beske-Diehl and Julie Bowles for paleomagnetic analysis, as well as Paul Wilkonson, Robert Flett and Dan Engstrom for ²¹⁰Pb analysis. The National Lacustrine Core Facility (LacCore) provided accommodation for core splitting, initial core description and sampling. This work is the result of research sponsored by the Minnesota Sea Grant College Program, supported by the NOAA National Sea Grant office, United States Department of Commerce, under grant No. NA07OAR4170009 to R.E.H., T.C.J. and J.P.W. The U.S. Government is authorized to reproduce and distribute reprints for government purposes, not withstanding any copyright notation that may appear hereon. This paper is journal reprint No. JR639 of the Minnesota Sea Grant College Program. Additional support for this research was provided by the National Science Foundation (NSF), grant No. OCE 0961720 to S.K., R.E.H. and J.P.W., and U.S. Environmental Protection Agency Cooperative Agreement GL-00E00790-2 to E.D.R. This document has not been subjected to the Agency's required peer and policy review and therefore does not necessarily reflect the view of the Agency, and no official endorsement should be inferred.

## Author contributions

M.D.O., J.P.W., R.E.H. and T.C.J. conceived the study; J.P.W., R.E.H., T.C.J., S.K. and E.D.R collected sediment cores and/or provided samples; M.D.O. processed samples, performed EA-IRMS analysis and analysed the data; S.K. performed diagenetic modelling; M.D.O. and J.P.W. wrote the paper. All authors discussed the results and commented on the manuscript.

## Additional information

**Competing interests:** The authors declare no competing financial interests.

