## [Peer Review File · Nature Communications]

Reviewers' Comments:

Reviewer #1:

Remarks to the Author:

Lines 25-27 - Somewhat misleading statement on effects of anthropogenic change on L. Superior when in Line 112 reference is made to papers (eg#10) detailing increased terrestrial inputs, including nutrient inputs.

Line 64 – I think you give a plausible mechanism for the changes and discuss why other mechanisms might be dismissed but the assertion that the changes can be **solely** explained by warming isn't warranted and I feel you need to be more circumspect.

Line 73-5 - Missing word 'ratios fall around 7'....7 what ?

Line 107 - Co-limitation of Fe and P needs a reference...and this is present day co-limitation so best to say 'Lake Superior is presently co-limited by Fe and P.

Line 128 - Reference 17 suggested that there are subtle changes Lake Superior with an increase in small-celled blue-green algae in spring and a recent decline in summer centric diatoms (possibly a result of lake warming and changes in water quality).

Line 161 - Explain what is an 'increase in export productivity'

Line 175 - Authors argue marked increased temperatures in the last 30 years (since c. 1979) but in a recent paper van Cleave et al *Limnol. Oceanogr.*, 59(6), 2014, 1889–1898 argue that there is a tipping point in Lake Superior "following the warm El Nino winter of 1997–1998, resulting in a "regime shift" in summer evaporation rate, water temperature, and numerous metrics of winter ice cover".

I mention this really to highlight that the authors have made no attempt to correlate in a statistical way the isotopic and climate records (which are various).

General Comment – would atmospheric Nitrogen deposition be another possible 'driver' of change with known increases over this late 20th C timescale ? Would be worth a sentence to discuss.

Reviewer #2:

Remarks to the Author:

Reviewer: Fred J Longstaffe

First, let me offer my apologies to the authors and editorial staff for an unavoidable delay in reading this interesting paper.

I have both the benefit of the latest version of the manuscript and the thorough previous reviews that it has received. I find the elemental and isotopic data on which the discussion is based to have been properly executed, and thoroughly representative of Lake Superior, especially for the last 200-year period, which is the focus of the manuscript. The efforts to achieve viable age-depth models for each of the cores are solid, and good use has been made of alternate methods given the paucity of radiocarbon suitable material.

As one of the authors of the 2011 study on the paleoproductivity of Lake Superior, I can confirm that the present study - in its focus on recent sediments - goes well beyond what was conducted in the earlier work. These authors have taken the next step that I would have taken - give suitable core materials - to drive the work into the relevant period of the Anthropocene, and done an excellent job in the process.

Key to the Discussion is the cause of the carbon-13 enrichment. In the latest version of the manuscript (the only one which I have seen), I think that one elephant in the room - diagenesis - is adequately addressed by acknowledging the consequences for TOC and providing literature to support the claims for a lack of carbon isotopic composition effect. A more compelling case could be made, but that would require compound-specific isotopic data.

Likewise, I am not entirely convinced that allochthonous contributions of C3 land plant-derived

organic matter can be ruled out entirely on the basis of C/N ratio alone (as suggested by this manuscript's Fig. 2D). Organic matter can be delivered to large-lake sediments on the surfaces of clay minerals, especially soil clays. Such clay-delivered organic matter can be characterized by low C/N, as has been noted by many previous researchers (e.g., see discussions for Lake Superior and Lake Ontario by Hyodo & Longstaffe, 2011, *Quat. Sci. Rev.*, and Hladyniuk & Longstaffe, 2015, *PPP*, for Lakes Superior and Ontario, respectively). The recalcitrant nature of clay mineral-fixed OM can lead to its selective preservation in lacustrine sediments, particular those characterized by low TOC. But that is a discussion for another day, and – with regard to the manuscript under consideration – would require its own set of special arguments to explain the similarity on patterns across the Superior Basin for so many cores.

I think this paper makes a very important point about anthropogenically increasing PP in yet another so-called pristine lake environment. It is an important 'Large Lake' companion to other recent wake-up calls of a similar nature for "pristine" arctic and high altitude (hyper) oligotrophic small lakes (e.g., Hundey et al., 2014, *Limnol. Oceanograph*; Hundey et al., 2016, *Nat. Comm.*). I recommend publication.

REVIEWERS' COMMENTS:

Reviewer #1 (Remarks to the Author):

Lines 25-27 - Somewhat misleading statement on effects of anthropogenic change on L. Superior when in Line 112 reference is made to papers (eg#10) detailing increased terrestrial inputs, including nutrient inputs.

We have put in additional wording, i.e. “The effects of anthropogenic changes in Lake Superior, which is Earth’s largest freshwater lake by area, are not well documented (spatially or temporally) and predicted future states in response to climate change vary”, in order to clarify that previous studies are limited to few specific sites that may not represent the entire lake.

Indeed, the data from the reference in question (reference 13; originally 10) – O’Beirne et al 2015) were not representative of the whole lake, nor of the Holocene. That paper focused only on two cores taken from the same site in the westernmost portion of the lake, just outside the Duluth harbor, and only on the last ~200 years. Furthermore, the earlier study focused on localized impacts related to nutrient inputs that did not impact the entire basin but were evident in the westernmost part of the basin in the middle of the 20th century (and subsequently reduced through implementation of water treatment plants in Duluth, MN).

Line 64 – I think you give a plausible mechanism for the changes and discuss why other mechanisms might be dismissed but the assertion that the changes can be solely explained by warming isn’t warranted and I feel you need to be more circumspect.

We have removed the use of “solely” and rephrased the sentence to read: “...which also demonstrates that abrupt increases in PP within the last century can be reasonably explained by the effects of anthropogenic climate warming.”

Line 73-5 - Missing word ‘ratios fall around 7’7 what ?

We have changed the wording to clarify that the average value of our $C_{org}:N$ data is seven which is indicative of algal sourced OM. Algae typically have atomic $C_{org}:N$ ratios between 4 and 10 as shown in reference 7.

“Values of atomic $C_{org}:N$ ratios in the cores presented are less than 10 (Fig. 2D), which indicates sediments dominated by algal sourced organic matter (OM).”

Line 107 - Co-limitation of Fe and P needs a reference...and this is present day co-limitation so best to say ‘Lake Superior is presently co-limited by Fe and P.

We have changed the wording to: “Lake Superior is presently co-limited by Fe and P.”

and referenced:

*Ivanikova, N.V., McKay, R.M.L., Bullerjahn, G.S. & Sterner, R.W. Nitrate utilization by phytoplankton in Lake Superior is impaired by low nutrient (P, Fe) availability and seasonal light limitation—a cyanobacterial bioreporter study. *J. Phycol.* 43, 475–484 (2007).*

*Sterner, R. W. et al. Phosphorus and trace metal limitation of algae and bacteria in Lake Superior. *Limnol. Oceanogr.* 49, 495– 507 (2004).*

Line 128 - Reference 17 suggested that there are subtle changes Lake Superior with an increase in small-celled blue-green algae in spring and a recent decline in summer centric diatoms (possibly a result of lake warming and changes in water quality).

In recognition of the subtle changes we have added the following after LINE 144:

“Furthermore, the subtle changes in algal divisions that have been documented are likely the result of warming water temperatures and accompanied by changes in water quality (reference 21).”

Line 161 - Explain what is an ‘increase in export productivity’

We have defined this by saying: “...increases in export production (i.e., an increase in the amount of organic matter produced by primary production that is not recycled) ...”

Line 175 - Authors argue marked increased temperatures in the last 30 years (since c. 1979) but in a recent paper van Cleave et al *Limnol. Oceanogr.*, 59(6), 2014, 1889–1898 argue that there is a tipping point in Lake Superior “following the warm El Nino winter of 1997–1998, resulting in a “regime shift” in summer evaporation rate, water temperature, and numerous metrics of winter ice cover”.

I mention this really to highlight that the authors have made no attempt to correlate in a statistical way the isotopic and climate records (which are various).

While we agree that the reviewer has a point, because the sample size in the time period of overlap between our cores and instrumental records is small, we cannot correlate these events with statistical certainty. Specifically, the resolution in the multicores, while still high resolution (every 0.5 cm or every 1 cm depending on the core), amounts to between 4 and 8 data points per core that encompass the same time period as reliable instrumental records (ca.1975 – present). The sample size simply isn’t large enough for robust statistical analysis. Our records still show a verifiable abrupt increase in proxy data coincident with the effects of anthropogenic climate warming in Lake Superior, despite the lack of statistical correlation.

General Comment – would atmospheric Nitrogen deposition be another possible ‘driver’ of change with known increases over this late 20th C timescale? Would be worth a sentence to discuss.

That is a possibility; however, nitrate concentrations exhibit a continuous build-up throughout the last century (Sterner et al. 2007) and to the present which would suggest that the concentration of nitrate/bioavailable nitrogen in the lake isn’t what has/is limiting primary production within the lake.

We have added the following after LINE 116:

“Increases in nitrate concentrations likely helped to fuel initial PP in the lake; however, the continuous build-up throughout the last century and extremely high N:P ratios at present (Sterner et al. 2007) suggest that the amount of available N is not what is limiting PP in Lake Superior.”

And referenced:

*Sterner, R.W. et al. Increasing stoichiometric imbalance in North America's largest lake: nitrification in Lake Superior. *Geophys. Res. Lett.* **34**, L10406 (2007)*

Reviewer #2 (Remarks to the Author):

Reviewer: Fred J Longstaffe

First, let me offer my apologies to the authors and editorial staff for an unavoidable delay in reading this interesting paper.

I have both the benefit of the latest version of the manuscript and the thorough previous reviews that it has received. I find the elemental and isotopic data on which the discussion is based to have been properly executed, and thoroughly representative of Lake Superior, especially for the last 200-year period, which is the focus of the manuscript. The efforts to achieve viable age-depth models for each of the cores are solid, and good use has been made of alternate methods given the paucity of radiocarbon suitable material.

As one of the authors of the 2011 study on the paleoproductivity of Lake Superior, I can confirm that the present study - in its focus on recent sediments - goes well beyond what was conducted in the earlier work. These authors have taken the next step that I would have taken – give suitable core materials – to drive the work into the relevant period of the Anthropocene, and done an excellent job in the process.

Key to the Discussion is the cause of the carbon-13 enrichment. In the latest version of the manuscript (the only one which I have seen), I think that one elephant in the room – diagenesis – is adequately addressed by acknowledging the consequences for TOC and providing literature to support the claims for a lack of carbon isotopic composition effect. A more compelling case could be made, but that would require compound-specific isotopic data.

Likewise, I am not entirely convinced that allochthonous contributions of C₃ land plant-derived organic matter can be ruled out entirely on the basis of C/N ratio alone (as suggested by this manuscript's Fig. 2D). Organic matter can be delivered to large-lake sediments on the surfaces of clay minerals, especially soil clays. Such clay-delivered organic matter can be characterized by low C/N, as has been noted by many previous researchers (e.g., see discussions for Lake Superior and Lake Ontario by Hyodo & Longstaffe, 2011, *Quat. Sci. Rev.*, and Hladyniuk & Longstaffe, 2015, *PPP*, for Lakes Superior and Ontario, respectively). The recalcitrant nature of clay mineral-fixed OM can lead to its selective preservation in lacustrine sediments, particular those characterized by low TOC. But that is a discussion for another day, and – with regard to the manuscript under consideration – would require its own set of special arguments to explain the similarity on patterns across the Superior Basin for so many cores.

In recognition that terrigenous organic matter adsorbed on clay particles could possibly influence C/N ratios we have added the following after LINE 81:

“Although soil OM adsorbed onto fine-grained clays can also exhibit C_{org}:N ratios <10 (Prahl et al., 1994; Batjes, 1996), the remarkable coherence among cores regardless of proximity to shore (the source of clay-derived OM) is more suggestive of algal sourced OM buried in the sediments.”

And referenced:

Prahl, F.G., Ertel, J.R., Goni, M.A., Sparrow, M.A., Eversmeyer, B. Terrestrial organic contributions to sediments on the Washington organic margin. Geochim. Cosmochim. Acta 58, 3035-3048 (1994).

Batjes, N.H. Total carbon and nitrogen in the soils of the world. Eur. J. Soil Sci. 47, 151-163 (1996).

I think this paper makes a very important point about anthropogenically increasing PP in yet another so-called pristine lake environment. It is an important 'Large Lake' companion to other recent wake-up calls of a similar nature for “pristine” arctic and high altitude (hyper) oligotrophic small lakes (e.g., Hundey et al., 2014, *Limnol. Oceanograph*; Hundey et al., 2016, *Nat. Comm.*). I recommend publication.